# CoD-Lite: Real-Time Diffusion-Based Generative Image Compression

**Zhaoyang Jia**[1][*]  **Naifu Xue**[2][*]  **Zihan Zheng**[1][*]  **Jiahao Li**[3]  **Bin Li**[3]
**Xiaoyi Zhang**[3]  **Zongyu Guo**[3]  **Yuan Zhang**[2]  **Houqiang Li**[1]  **Yan Lu**[3]

## Abstract

Recent advanced diffusion methods typically derive strong generative priors by scaling diffusion transformers. However, scaling fails to generalize when adapted for real-time compression scenarios that demand lightweight models. In this paper, we explore the design of real-time and lightweight diffusion codecs by addressing two pivotal questions. **First, does diffusion pre-training benefit lightweight diffusion codecs?** Through systematic analysis, we find that generation-oriented pre-training is less effective at small model scales whereas compression-oriented pre-training yields consistently better performance. **Second, are transformers essential?** We find that while global attention is crucial for standard generation, lightweight convolutions suffice for compression-oriented diffusion when paired with distillation. Guided by these findings, we establish a one-step lightweight convolution diffusion codec that achieves real-time 60 FPS encoding and 42 FPS decoding at 1080p. Further enhanced by distillation and adversarial learning, the proposed codec reduces bitrate by 85% at a comparable FID to MS-ILLM, bridging the gap between generative compression and practical real-time deployment. Codes are released at https://github.com/microsoft/GenCodec/tree/main/CoD_Lite.

## 1. Introduction

The deployment of neural image codecs (Ballé et al., 2017; Ballé et al., 2018) is governed by two key constraints: perceptual fidelity and inference latency. Ideally, a codec should deliver photorealistic reconstructions at speeds suit-

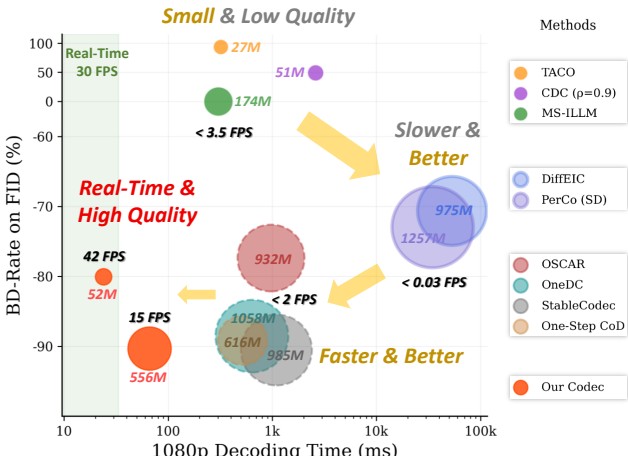

*Figure 1.* Progress in generative image codecs has largely been driven by scaling models, incurring substantial decoding latency. In contrast, our codec achieves a superior trade-off between perceptual quality and coding speed, enabling real-time 1080p decoding on an A100 GPU while attaining near state-of-the-art FID. Decoding parameters and time are shown.

able for real-time applications. However, recent advances in generative compression (Mentzer et al., 2020; Careil et al., 2024) have largely driven these objectives apart.

To transcend the perceptual limits inherent in distortion-based optimization (Blau & Michaeli, 2019), generative compression leverages generative priors to synthesize high-frequency details. However, this pursuit has become entangled with aggressive model scaling in recent advancements, as shown in Figure 1. While distortion-optimized codecs like ELIC (He et al., 2022) typically use fewer than 10M parameters, early generative approaches such as MS-ILLM (Muckley et al., 2023) already exceed 100M. More recently, diffusion-based codecs like PerCo (Careil et al., 2024) have pushed this trend further, relying on billion-parameter foundation models (e.g., Stable Diffusion (Rombach et al., 2022)) to ensure generation quality.

Although such large-scale models achieve impressive visual realism, they incur prohibitive decoding latency even exceeding 10 seconds. Even with advances like one-step diffusion, state-of-the-art systems such as StableCodec (Zhang et al., 2025) operate below 3 FPS, failing to meet real-time

---

* This work was done when Zhaoyang Jia, Naifu Xue and Zihan Zheng were full-time interns at Microsoft Research Asia. [1]University of Science and Technology of China [2]Communication University of China [3]Microsoft Research Asia.

requirements. Consequently, diffusion-based codecs remain largely impractical for latency-sensitive applications.

In this work, we challenge the prevailing *scaling-up* paradigm, revisiting diffusion-based compression from an efficiency-centric perspective. Drawing inspiration from efficiency efforts like TinySR (Dong et al., 2025), we investigate whether perceptual quality and real-time performance can be reconciled through principled architectural and training designs for lightweight diffusion models. Specifically, we explore it by addressing two key questions.

**First, does diffusion pre-training benefit lightweight diffusion codecs?** While diffusion pre-training is critical for improving large diffusion codecs, its effectiveness in the lightweight regime remains unclear. Through a systematic study, we reveal that: while generation-oriented pre-training substantially improves large models (700M), it offers negligible gains for lightweight ones (34M). We attribute this to a difficulty–capacity mismatch, where synthesizing rich visual content from extremely sparse semantic signals (i.e., class labels or text prompts) places demands beyond the representational capacity of lightweight diffusion backbones.

A natural remedy is to provide more informative conditioning. To achieve this, we leverage compression-oriented pre-training (i.e., CoD (Jia et al., 2025b)) to learn image-native, information-dense conditions. This reduces the modeling burden and makes diffusion compatible with lightweight architectures even at 34M parameters, achieving much stronger perceptual performance when fine-tuned to a codec.

**Second, are transformers essential for compression-oriented diffusion?** Diffusion Transformers (DiTs) (Peebles & Xie, 2023; Vaswani et al., 2017) underpin state-of-the-art diffusion models including CoD. However, their quadratic $O(N^2)$ attention complexity poses a major obstacle to real-time compression. Although global attention is indispensable for synthesizing coherent structures in generation, its necessity in compression remains an open question.

By analyzing attention patterns in CoD, we observe that attention collapses to local neighborhoods across most layers, as the compressed conditions already encode global structure. This shift from global to local modeling renders long-range attention largely redundant. Consequently, lightweight convolutional backbones, with inherent local inductive biases, are sufficient to capture the high-frequency textures required for compression.

Guided by these insights, we introduce a **real-time and lightweight one-step convolution diffusion image codec**. Built on compression-oriented diffusion pre-training and an efficient depth-wise convolutional backbone, the model is further distilled into one-step under a unified distillation and adversarial training framework. Our codec strikes a strong balance between perceptual fidelity and coding la-

tency. It employs a compact 28M encoder and 52M decoder to support real-time deployment with 60 FPS encoding and 42 FPS decoding at 1080p, and achieves an 85% bitrate reduction at comparable FID relative to MS-ILLM.

Our contributions are summarized as follows:

- We reveal compression-oriented diffusion pre-training as uniquely effective for lightweight diffusion codecs.

- We show that global attention can be replaced by convolutions in compression-oriented diffusion, with minimal loss and substantially faster speed.

- We propose a real-time diffusion codec that achieves 85% bits saving at comparable FID to MS-ILLM, while enabling low-latency 42 FPS decoding at 1080p.

## 2. Related Works

**Neural Image Compression.** Traditional neural image compression (NIC) optimizes rate-distortion performance using autoencoders (Ballé et al., 2017). While advancements (Cheng et al., 2020) achieve high PSNR, they often suffer from blurry reconstructions at low bitrates due to the pixel-wise distortion metrics. To improve perceptual quality, generative adversarial networks (GANs) (Goodfellow et al., 2020) have been integrated into compression frameworks (Agustsson et al., 2019; Mentzer et al., 2020; Lee et al., 2024; Jia et al., 2024; Körber et al., 2024; Agustsson et al., 2023) for synthesizing realistic textures.

**Diffusion-based Compression.** Diffusion models (Ho et al., 2020; Chen et al., 2024; Stability AI, 2023) have recently surpassed GANs in generation quality. Early diffusion codecs (Lei et al., 2023; Ke et al., 2025; Li et al., 2024; Theis et al., 2022; Elata et al., 2025; Xu et al., 2024) employed multi-step sampling, achieving superior perceptual fidelity but incurring prohibitive latency. They typically leverage large-scale foundation models as priors, further increasing computational cost. To accelerate inference, one-step diffusion codecs (Guo et al., 2025; Zhang et al., 2025; Xue et al., 2025) have been proposed. However, these methods typically rely on heavy backbones (e.g., DiT (Peebles & Xie, 2023) or UNet), limiting their real-time applicability. Recently, CoD (Jia et al., 2025b) proposed compression-oriented pre-training, offering the flexibility to customize diffusion foundation models directly for compression.

**Real-Time Neural Compression.** Real-time capability is essential for practical media applications. In the realm of image coding, architectures such as ELIC (He et al., 2022) and standards like EVC have been optimized for high efficiency. Similarly, neural video coding has seen significant progress towards real-time processing with methods like DCVC-RT (Jia et al., 2025a). However, real-time

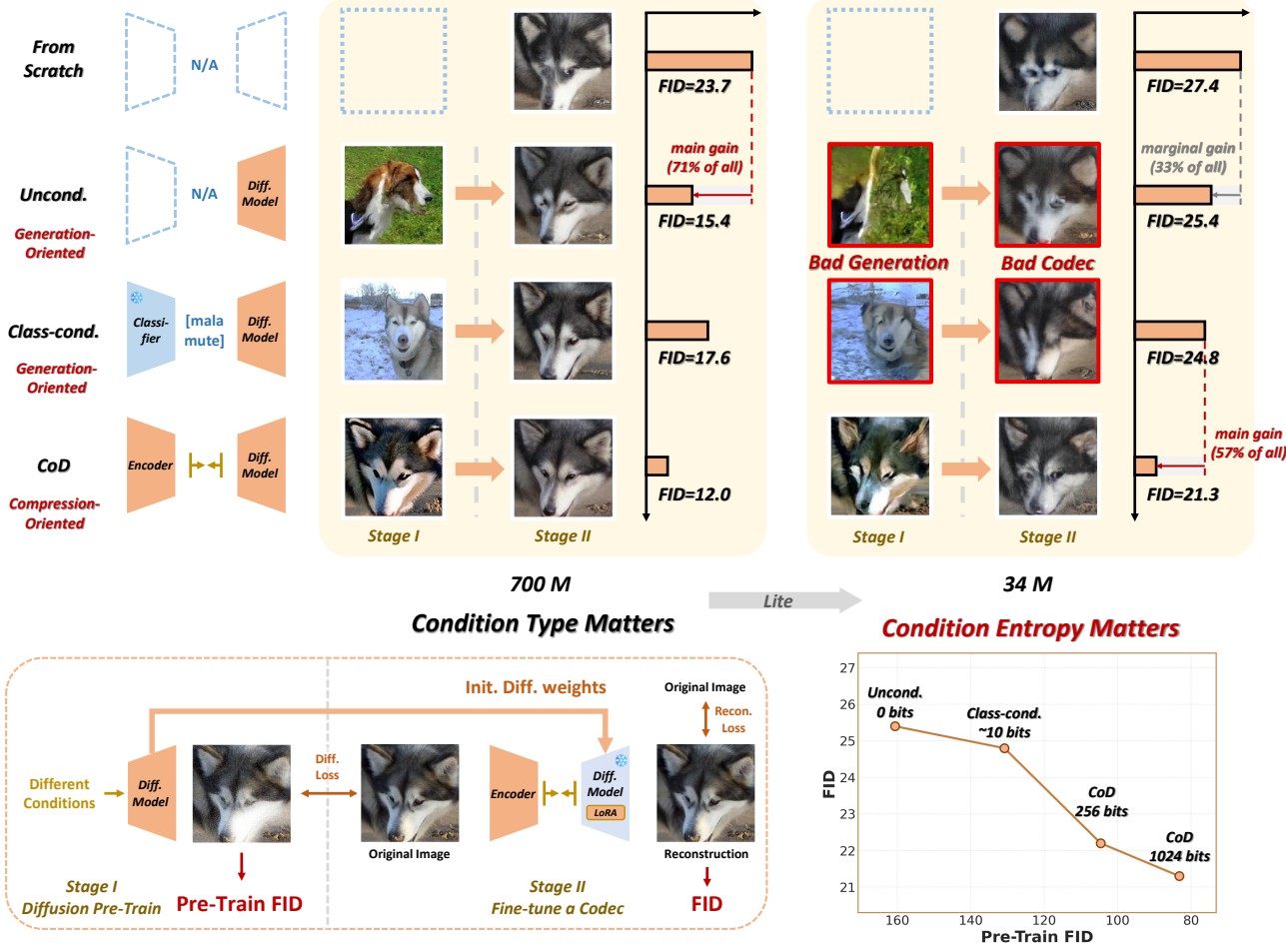

Figure 2. Analysis on diffusion pre-training at different model scales. The codecs target $0.0156$ bpp on $256 \times 256$ images.

performance in generative compression, particularly with diffusion-based models, remains largely unexplored due to the prohibitive computational costs of existing backbones.

## 3. Analysis: Diffusion Pre-training at Scale

Modern diffusion-based codecs derive superior perceptual quality from the rich generative priors of large-scale, pre-trained foundation models. However, the resulting prohibitive computational cost prevents their use in real-time applications. To bridge the gap, it is essential to scale down the models. This raises a fundamental question: *Does diffusion pre-training benefit the lightweight regime?* We investigate this question through a systematic empirical study.

### 3.1. Experimental Setup

We adopt the advanced pixel-space diffusion backbone PixNerd (Wang et al., 2025a) and train it on ImageNet (Russakovsky et al., 2015) at a resolution of $256 \times 256$. We train two sets of diffusion codecs at representative capacities of

700M and 34M using a two-stages manner. The encoder-decoder framework follows CoD (Jia et al., 2025b), where the encoder applies $16\times$ spatial downsampling. The vector quantization bottleneck employs a 4-bit codebook ($2^4 = 16$ codes), yielding an overall bitrate of $0.0156$ bpp.

In **Stage I**, the diffusion backbone is pre-trained via flow-matching loss. In **Stage II**, we adapt it into a one-step diffusion codec with $L_1$, LPIPS (Zhang et al., 2018) and PatchGAN (Demir & Unal, 2018) adversarial loss. To preserve generative priors, we update the diffusion decoder via LoRA (Hu et al., 2022). In addition, we also train codecs from scratch to serve as baselines. After training, we evaluate reconstruction quality using FID on $1,000$ images from the ImageNet validation set. Additional details are provided in Appendix A.1.

### 3.2. Disparity at Different Scales

Figure 2 illustrates a significant divergence in how diffusion pre-training scales across different model capacities.

**Unconditional Generation-oriented Pre-training**. It is effective at scale but limited for small models. For the large 700M model, it substantially reduces FID from 23.7 to 15.4. This trend reverses in the small 34M model, where modeling complex image distributions exceeds the model's representational limits. The pre-training fails to yield high quality samples, which propagates to downstream coding: fine-tuning improves FID by only 2.0 over random initialization, in stark contrast to the 8.3 gain in the large model.

**Class-conditioned Generation-oriented Pre-training**. It exhibits opposite effects across scales, benefiting small-capacity models more than large-capacity ones. For large models, prior work has shown that text-based conditioning can be detrimental to compression-oriented objectives (vonderfecht & Liu, 2025; Jia et al., 2025b). Consistent with these findings, we observe that class conditions provide little benefit and underperform unconditional pre-training. When model capacity is sufficient to capture the image distribution, part of capacity is diverted toward modeling image–label correlations that are largely irrelevant for compression. Consequently, these parameters transfer poorly during codec fine-tuning, leading to inferior performance.

In contrast, the conclusion shifts for small-capacity models, where representational power is insufficient to model the full image distribution. From an information-theoretic perspective, class conditioning supplies approximately 10 bits of side information ($2^{10} \approx 1000$ classes) to the diffusion model during reconstruction. This additional information effectively reduces the entropy of the target distribution, alleviating the capacity bottleneck. As a result, class-conditioned pre-training outperforms unconditional pre-training, but the improvement is limited by only 10 bits of conditions.

**Compression-oriented Pre-training**. This new perspective inspires us to enhance diffusion pre-training by injecting more informative conditions. This aligns naturally with the philosophy of CoD, which learns conditions carrying substantially more information (e.g., 1024 bits). Compared to generation-oriented pre-training, CoD yields substantial improvements, reducing FID by 3.5 for small models, comparable to the 3.4 gain in large-capacity models.

The experiments reveal a shift in the governing factors of diffusion pre-training across scales. At 700M parameters, sufficient model capacity allows unconditional pre-training to perform well, with the condition type further determining performance. At 34M parameters, limited capacity makes the entropy of condition information the dominant factor. As shown in Figure 2, the number of conditioning bits is strongly correlated with pre-training quality, which in turn directly influences downstream codec performance.

The takeaways of the above analysis are the following:

> **Takeaways for Diffusion Pre-train at Scales:**
>
> **1. Pre-training efficacy depends on scale.** Unconditional pre-training benefits large models but degrades in the lightweight regime. Class-conditioned pre-training helps small models overcome capacity limits, while offering limited benefit for large models.
>
> **2. Condition entropy matters in low-capacity regimes.** For small models, performance strongly correlates with the entropy of conditioning information. High-information conditions of CoD significantly outperform standard generation-oriented pre-training.

## 4. Analysis: Diffusion Transformers in CoD

Recently, Diffusion Transformers (DiTs) have become the backbone of advanced generative models and codecs (vonderfecht & Liu, 2025; Jia et al., 2025b). However, the $\mathcal{O}(N^2)$ complexity of global attention poses a major barrier to real-time deployment: even a small 34M PixNerd model requires approximately 300 ms to decode a 1080p image.

In contrast to generation that synthesizes global structure from scratch, compression operates on rich representations that already preserve global layout and primarily focuses on generating local details. This observation raises a critical question: *Is global attention truly necessary for CoD?*

### 4.1. Visualizing the Attention Landscape

To investigate this, we visualize the attention maps of a CoD model across all 26 layers in Figure 3. The results are striking: only 7 layers exhibit global receptive fields, while the remaining 19 attend exclusively to local neighborhoods, revealing a clear two-phase attention pattern.

**Alignment-Induced Aggregation (Layers 0–7).** In the shallow layers, attention appears to expand from local to global. However, closer inspection reveals that this behavior is primarily induced by REPA (Yu et al., 2025) feature alignment, which enforces correspondence with DINOv2 (Oquab et al., 2024) features at Layer 7. Supporting evidence shows that attention disproportionately focuses on sink tokens (e.g., top-left positions) at this stage. Masking these tokens results in negligible quality degradation, indicating that the observed global attention mainly serves alignment objectives rather than essential generative modeling.

**Focused Structure Refinement (Layers 8–25).** Following REPA alignment, attention rapidly collapses to a predominantly local focus. While a few layers still capture long-range semantic dependencies, most attention mass concentrates within local neighborhoods.

**1. DiT-based Compression-oriented Diffusion (CoD) Model**

**2. DiT Attention map visualization**

**3. Statistical analysis**

*(a) Where does the attention focus?*   *(b) How does the attention evolve across blocks?*   *(c) Across timesteps?*

*Figure 3.* Analysis on DiT in compression-oriented diffusion models. More visualizations and illustrations are in Appendix A.2.

| Multi-Step Pre-Train @ 0.0039 bpp | FID |
|---|---|
| DiT → Global Attn. | 59.7 |
| + Local Window Attn. | 60.2 |
| + Replace Attn. by Conv. | 62.5 |

| One-Step Fine-Tune @ 0.0312 bpp | Diff. Param. | Dec. Speed | DISTS | FID |
|---|---|---|---|---|
| DiT → Global Attn. | 44 M | 331.2 ms | 0.118 | 37.5 |
| w/ DMD Distillation loss | | | 0.118 | 35.4 |
| + Replace Attn. by Conv. | 40 M | 23.6 ms | 0.115 | 41.4 |
| w/ DMD Distillation loss | | | 0.114 | 36.4 |

*Table 1.* Ablation study on DiT. Left: Pre-trained multi-step diffusion foundation model. Right: Fine-tuned one-step diffusion codec.

Statistical analysis in Figure 3 about average attention distance confirms this trend, revealing that attention mass is concentrated locally with a negligible long-range tail. Furthermore, as the noise level decreases (typically corresponding to higher bitrates in compression (Guo et al., 2025)), this localization becomes increasingly pronounced.

### 4.2. From Global Attention to Local Convolution

The dominance of local interactions suggests that costly global attention can be substituted with efficient local operators. We validate this via an ablation study on CoD (Table 1),

measured on Kodak (Eastman Kodak Company, 1999).

**Pre-training: Local Operators are enough.** We first compare global attention with local operators in CoD pre-training. Local window attention achieves performance on par with global attention and lightweight depth-wise convolutions incur only a modest degradation, confirming that explicit global context is not necessary.

**Fine-Tuning: Convolutions Match Transformers via Distillation.** When fine-tuned as a one-step codec, the convolution backbone achieves a $14\times$ speedup at the cost of FID

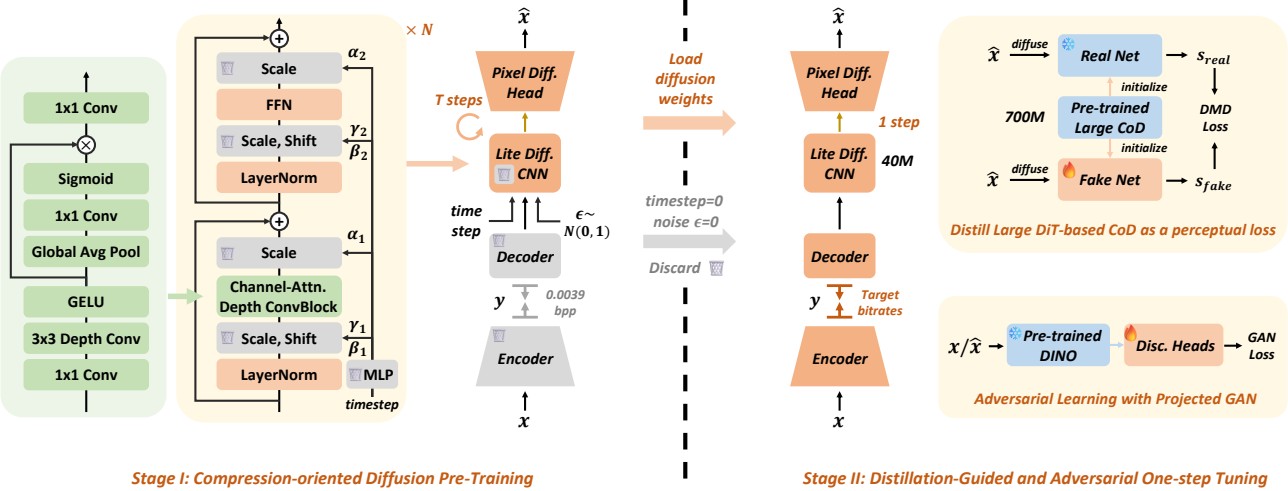

*Figure 4.* Framework overview of proposed real-time diffusion based image codec.

degradation (41.4 vs. 37.5). We attribute this gap primarily to the optimization difficulty of depth-wise convolution networks rather than limited representational capacity. By introducing the DMD distillation loss (Section 5), this gap is largely closed (36.4 vs. 35.4). These results demonstrate that efficient convolution backbones can match DiT performance within the CoD framework under proper training, enabling true real-time diffusion compression.

The takeaways of the above analysis are the following:

> **Takeaways for Diffusion Transformers in CoD:**
>
> **1. Global attention is redundant for CoD.** Most attentions focus on local interactions, rendering the costly global attention unnecessary.
>
> **2. Convolution enables real-time and high performance.** Distilled convolution diffusion can achieve comparable quality to DiTs with a $14\times$ speedup, provided that a DiT-based teacher is used for distillation.

# 5. Real-Time Diffusion-Based Compression

Leveraging the insights from previous sections, we propose a real-time diffusion-based codec, as illustrated in Figure 4.

## 5.1. Framework

The proposed codec uses an encoder and decoder to compress the conditions, which guide a lightweight one-step convolutional diffusion module with a decoupled diffusion head (Wang et al., 2025b) for direct pixel reconstruction.

**Encoder, Entropy, and Decoder**. Following CoD, we build the encoder and decoder using residual blocks (He et al.,

2016), and constrain the bottleneck via vector quantization (Esser et al., 2021) with a learned codebook. We utilize fixed-length coding to encode the codebook indices. By varying the codebook size and the latent size, our codecs cover a wide bitrate range from 0.0039 to 0.5 bpp.

**Lightweight Convolution Diffusion Module**. The diffusion backbone follows the design principles of DeCo (Ma et al., 2025), adopting a pixel-space DiT with $16 \times 16$ patch embedding and an MLP-based pixel head. To improve efficiency, we replace the computationally expensive attention modules with depth-wise convolution blocks augmented by channel attention (Hu et al., 2018; Ai et al., 2025), and substantially reduce both the channel width and the number of blocks, yielding a compact backbone with 52M parameters. Moreover, since AdaLN-Zero (Peebles & Xie, 2023) in CoD is solely used for timestep conditioning and becomes redundant in the one-step setting, we remove it to further reduce the backbone size to 40M parameters.

## 5.2. Training

We employ a two-stage training pipeline for one-step diffusion codecs: first pre-training the diffusion prior using CoD, and then fine-tuning the codec at specific bitrates.

**Stage I: Compression-Oriented Diffusion Pre-training**. In the first stage, we focus on learning a robust generative prior suitable for compression. Following CoD, we end-to-end learn a compression-oriented condition with a bitrate constraint of 0.0039 bpp, utilizing a unified flow matching loss (Jia et al., 2025b). The model uses $\mathcal{X}$ prediction following the success in pixel diffusion (Li & He, 2025).

**Stage II: Distillation-Guided and Adversarial One-Step Tuning**. We discard the stochastic sampling process and fix the timestep $t = 0$ and noise $\epsilon = 0$ to transform the

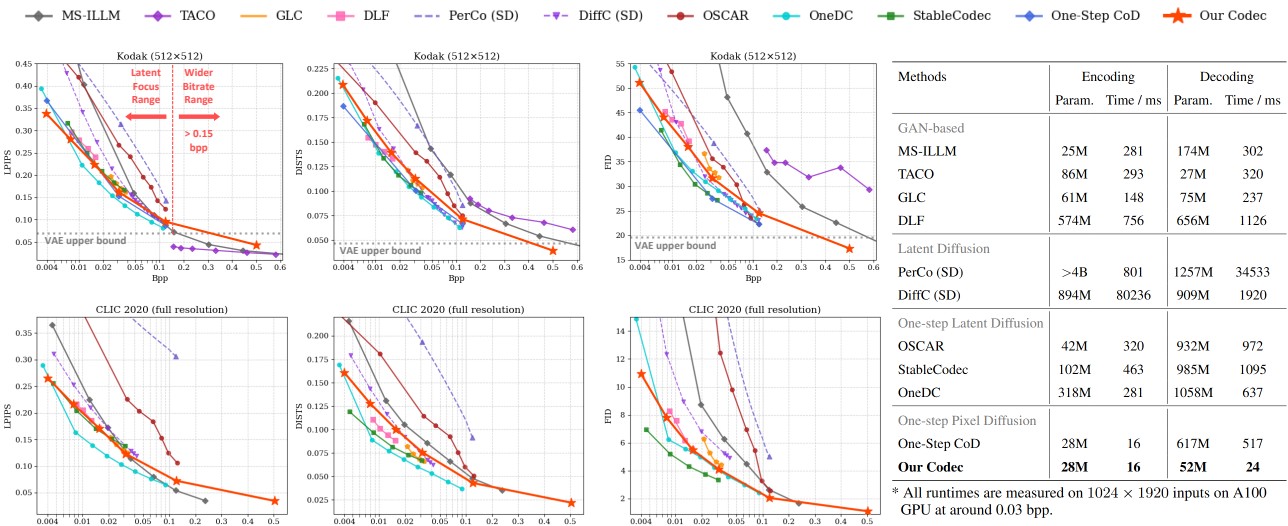

| Methods | Encoding | | Decoding | |
|---|---|---|---|---|
| | Param. | Time / ms | Param. | Time / ms |
| *GAN-based* | | | | |
| MS-ILLM | 25M | 281 | 174M | 302 |
| TACO | 86M | 293 | 27M | 320 |
| GLC | 61M | 148 | 75M | 237 |
| DLF | 574M | 756 | 656M | 1126 |
| *Latent Diffusion* | | | | |
| PerCo (SD) | >4B | 801 | 1257M | 34533 |
| DiffC (SD) | 894M | 80236 | 909M | 1920 |
| *One-step Latent Diffusion* | | | | |
| OSCAR | 42M | 320 | 932M | 972 |
| StableCodec | 102M | 463 | 985M | 1095 |
| OneDC | 318M | 281 | 1058M | 637 |
| *One-step Pixel Diffusion* | | | | |
| One-Step CoD | 28M | 16 | 617M | 517 |
| **Our Codec** | **28M** | **16** | **52M** | **24** |

\* All runtimes are measured on $1024 \times 1920$ inputs on A100 GPU at around 0.03 bpp.

*Figure 5.* Rate-distortion curves (left) and complexity analysis (right).

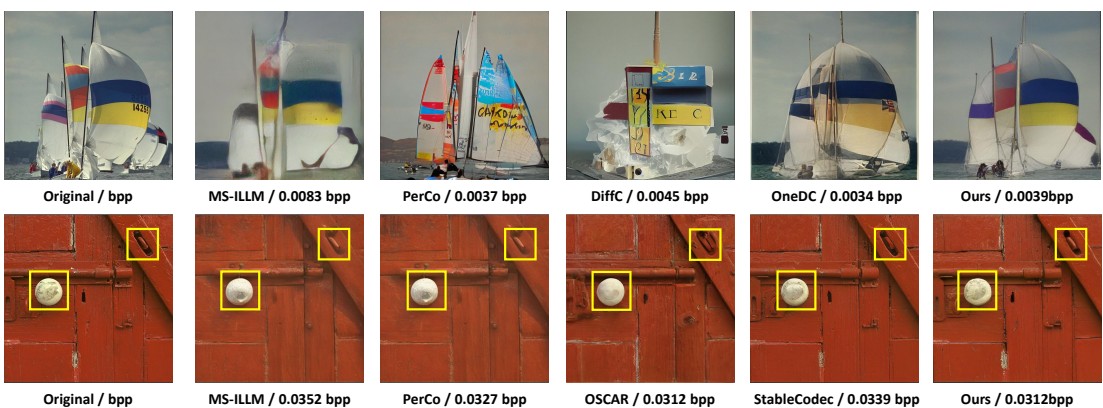

*Figure 6.* Visual comparison with baselines. More visual results are in Appendix B.3.

pre-trained diffusion model into a one-step deterministic generator. Beyond the reconstruction objective $L_1$, perceptual objective $L_P$ (Zhang et al., 2018), and codebook commitment loss $L_C$ (Esser et al., 2021), we enhance the model using distillation loss $L_{\text{DMD}}$ and adversarial loss $L_{\text{GAN}}$.

$$L = L_1 + L_P + \lambda_C \cdot L_C + \lambda_{\text{DMD}} \cdot L_{\text{DMD}} + \lambda_{\text{GAN}} \cdot L_{\text{GAN}} \quad (1)$$

**For distillation**, we use a pre-trained DiT-based CoD as the teacher and distill our codec following the scheme of Distribution Matching Distillation (DMD (Yin et al., 2024)). We adopt the pre-trained CoD (pixel space, 700M) to perform distillation directly in the pixel domain. The DMD loss estimates real and fake scores using the teacher to directly optimize the reconstruction toward real distribution. As in Section 4, although direct optimization of depth-wise convolution can cause performance degradation, DMD distillation significantly recovers performance, enabling a strong codec.

**For adversarial training**, we incorporate a projected GAN

loss (Sauer et al., 2021). Specifically, we employ a multi-scale discriminator that projects input images onto feature pyramids extracted from a fixed DINOv2 (Oquab et al., 2024) encoder, providing robust semantic guidance.

## 6. Experiments

### 6.1. Implementation Details

**Training**. We train our diffusion-based codec with 22M images from ImageNet-21K (Russakovsky et al., 2015), OpenImages (Kuznetsova et al., 2020), and SA-1B (Kirillov et al., 2023), at a resolution of up to $512 \times 512$. More detailed training settings are in Figure 12.

**Evaluation**. We benchmark performance on the Kodak dataset (Eastman Kodak Company, 1999) at center-cropped $512 \times 512$ resolution and the CLIC2020 test set (Toderici et al., 2020) at full resolution, utilizing LPIPS (Zhang

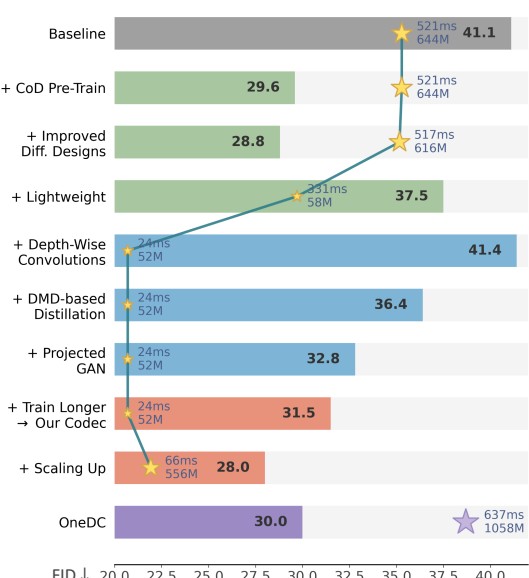

*Figure 7.* Ablation study via a roadmap.

| Kodak resolution | 128x128 | 256x256 | 512x512 |
|---|---|---|---|
| MS-ILLM | 0 | 0 | 0 |
| StableCodec | -31 / -38 | -44 / -70 | -36 / -77 |
| OneDC | -13 / -11 | -41 / -72 | **-54 / -78** |
| Ours | **-79 / -80** | **-62 / -75** | -40 / -70 |

*Table 2.* BD-Rate (%) calculated on LPIPS / DISTS on lower resolution version of Kodak.

et al., 2018), DISTS (Ding et al., 2020), and FID (Heusel et al., 2017) as evaluation metrics. FID measurement follows (Ohayon et al., 2025), using overlapped $64 \times 64$ patches on Kodak and $256 \times 256$ patches on CLIC. For coding time, we measure the latency on $1024 \times 1920$ images at around 0.03 bpp on a single NVIDIA A100 GPU. We compare against a comprehensive set of codecs, including: GAN-based codecs MS-ILLM (Muckley et al., 2023), TACO (Lee et al., 2024), and GLC (Jia et al., 2024); multi-step diffusion codecs PerCo (SD) (Körber et al., 2024) and DiffC (vonderfecht & Liu, 2025); one-step diffusion codecs OSCAR (Guo et al., 2025), StableCodec (Zhang et al., 2025), OneDC (Xue et al., 2025), and One-Step CoD (Jia et al., 2025b).

## 6.2. Results

As illustrated in Figure 5, our codec outperforms GAN-based and multi-step diffusion codecs across most metrics on Kodak. It also achieves competitive quality compared to advanced one-step diffusion codecs while delivering at least $20\times$ faster decoding speeds. Quantitatively, our method achieves approximately 85% bit savings on FID compared

| Enc. / Dec. Time (ms) | $720 \times 480$ | $1920 \times 1024$ | $3840 \times 2160$ |
|---|---|---|---|
| NVIDIA A100 GPU | 6 / 13 | 16 / 24 | 63 / 75 |
| NVIDIA RTX 4090 GPU | 4 / 10 | 17 / 21 | 75 / 88 |
| NVIDIA RTX 2080Ti GPU | 9 / 21 | 46 / 47 | 200 / 208 |
| AMD EPYC 9V84 Processor | 58 / 65 | 287 / 202 | 1343 / 837 |

*Table 3.* Coding speed test across different resolution and devices.

| Module | Encoder | Decoder | Lite Diff. CNN | Pixel Head |
|---|---|---|---|---|
| Parameters | 28.1 M | 10.9 M | 37.9 M | 3.2 M |
| kMACs / pixel | 325 | 43 | 138 | 29 |
| Runtime@$1024 \times 1920$ | 15.5 ms | 2.1 ms | 14.5 ms | 6.2 ms |

*Table 4.* Module-wise complexity break down on A100 GPU.

to MS-ILLM (measured with BD-rate (Bjontegaard, 2001)).

On CLIC at very high resolutions, our codec surpasses most baselines and rivals state-of-the-art methods, though it exhibits a slight performance drop on DISTS compared to that on low-resolution Kodak. We attribute this to our training resolution being limited to $512 \times 512$. We believe this can be addressed by training on higher-resolution in future work.

**Robustness on lower resolutions**. Although our codec exhibits a slight performance drop at higher resolutions due to a lack of high-resolution training, we find that it excels at lower resolutions. In Table 2, we evaluate the performance of generative codecs on the Kodak dataset under reduced resolutions. Previous diffusion-based codecs exhibit significant performance loss at lower resolutions. In contrast, our codec performs better as the resolution decreases, even though it was never trained on images as low as $128 \times 128$. This demonstrates the robust generalizability of our method to lower resolutions.

**Visual Comparison**. Figure 6 presents qualitative comparisons. At a low bitrate of approximately 0.03 bpp, our real-time codec achieves visual quality competitive with state-of-the-art heavyweight codecs, highlighting its strong potential for practical deployment. Figure 8 presents a visual comparison of our reconstruction on a high-resolution 2K image from the CLIC2020 test set. Although our codec was not trained on higher resolutions, it still delivers high-quality visual results even at an extremely low bitrate of 0.0039 bpp, with no observable blocking artifacts.

**Complexity Analysis**. We test coding speed on different GPU and CPU devices across different resolutions in Table 3. Results show the speed of our codec on consumer GPU, CPU and at ultra-high resolution. The break-down analysis of each module is in Table 4.

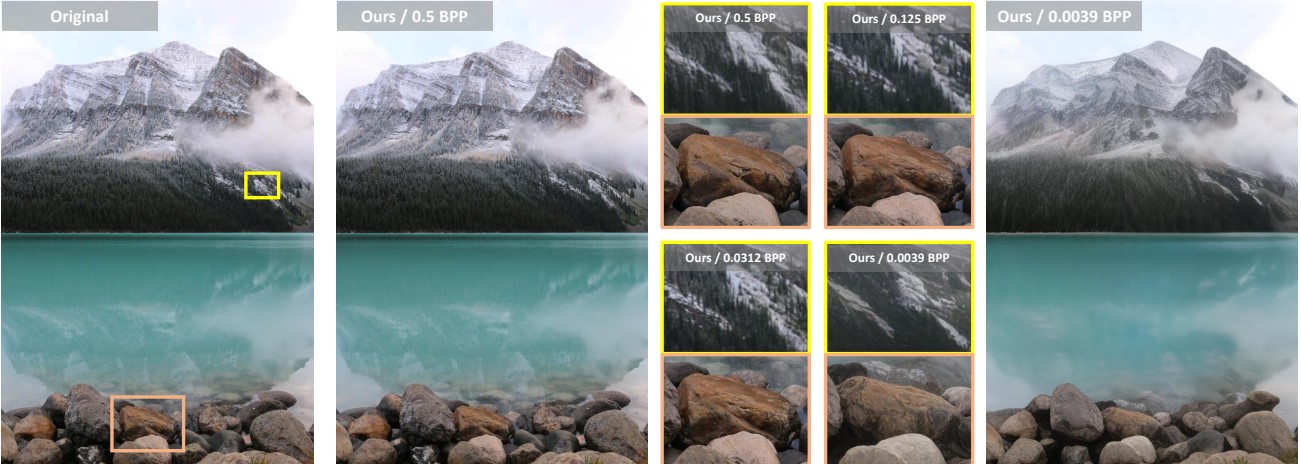

*Figure 8.* Visual results on high resolution 2K image.

### 6.3. Discussion: Advancing in a Wide Bitrate Range

Most existing diffusion-based codecs are constrained by the latent space of VAEs (Kingma & Welling, 2013) and primarily operate below 0.15 bpp. Pixel-space GAN-based codecs do not suffer from explicit bitrate limitations, but their performance at low bitrates is poor. In contrast, our codec provides a win-win solution: it scales to much higher bitrates (up to 0.5 bpp) while maintaining strong performance at ultra-low bitrates (like 0.0039 bpp).

### 6.4. Ablation: A Roadmap

In this section, we start with a baseline that is trained from-scratch with PatchGAN. Its model structure uses PixNerd following one-step CoD. We then demonstrate the incremental integration of each component to build a robust real-time codec. We report 1080p decoding speeds, decoding parameters, and FID on Kodak at 0.0312 bpp in Figure 7.

Our **Baseline** achieves an FID of 41.1 with a latency of 521 ms. We first perform **CoD pre-training**, following (Jia et al., 2025b) to pre-train it with $\mathcal{V}$-prediction (Salimans & Ho, 2022), significantly improving FID to 29.6. Then we introduce **Improved Diffusion Designs**, adopting the advanced pixel diffusion head from DeCo and pre-training with $\mathcal{X}$-prediction, which slightly boosts both performance and speed. Next, we significantly reduce diffusion parameters from 467M to 44M to create a **Lightweight** model. This increases FID to 37.5 but reduces latency to 331 ms. By replacing self-attention with **Depth-Wise Convolutions**, decoding speed is accelerated by $14\times$ to 24 ms, with an FID drop to 41.4. **DMD-based Distillation** significantly improves FID to 36.4, and replacing PatchGAN with **Projected GAN** further reduces it to 32.8. Finally, we **Train it Longer** to yield a final FID of 31.5 of **Our Codec**. To compare with state-of-the-art codecs, we **Scale Up** the model

parameters to 556M, demonstrating an FID of 28.0 while maintaining a fast decoding speed of 66 ms, which outperforms OneDC with $9.6\times$ speedup.

## 7. Conclusion

We introduced a real-time, lightweight convolutional diffusion-based image codec. Our analysis reveals that compression-oriented diffusion pre-training effectively enables lightweight models, and that global attention can be replaced by efficient convolutions without sacrificing quality in the compression context. The resulting codec achieves competitive FID with state-of-the-art methods while delivering real-time 1080p performance, marking a significant step towards practical generative image compression.

**Limitations.** Our current model is trained on $512 \times 512$ resolution, leading to reduced performance when scaling to very high resolutions (e.g., 4K). We plan to address high-resolution training in future work.

## Impact Statement

This paper presents a real-time diffusion-based image compression method. Our work improves the efficiency of digital media storage and transmission, with the potential to reduce bandwidth consumption and energy usage in data centers. However, as with other generative compression approaches, there is a risk of producing realistic but non-existent details (i.e., hallucinations), which may be unsuitable for applications requiring strict fidelity, such as medical imaging or forensics. We therefore encourage users to carefully consider application-specific requirements when deploying generative codecs. Outside of these specific cases, we believe our method will broadly benefit the community by democratizing high-quality, low-latency image coding.

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

# A. Experimental Details

This section provides comprehensive details of the experimental configurations presented in the paper. We organize the content according to the three main analyses: diffusion pre-training at scale (Section A.1), diffusion transformers in CoD (Section A.2), and the proposed real-time codec (Section A.3).

## A.1. Analysis on Diffusion Pre-training at Scale

This subsection provides implementation details for the experiments in Section 3, where we investigate the effectiveness of diffusion pre-training across different model scales.

**Model Architecture.** We adopt the PixNerd (Wang et al., 2025a) architecture for the diffusion module. For the **large-scale variant** (700M parameters), we configure the hidden dimension to 1152, with 26 DiT blocks and 4 decoupled pixel head blocks. For the **lightweight variant** (34M parameters), we reduce the hidden dimension to 384, with 10 DiT blocks and 4 decoupled pixel head blocks.

**Training Protocol.** Training is conducted on the ImageNet (Russakovsky et al., 2015) training set at $256 \times 256$ resolution using a two-stage approach:

*Stage I (Diffusion Pre-training)*: Following the training process of PixNerd and CoD, the diffusion backbone is pre-trained using flow-matching loss with $\mathcal{V}$-prediction. We train with a batch size of 64 for 800k steps (40 epochs total) using a learning rate of $10^{-4}$.

*Stage II (Codec Fine-tuning)*: The pre-trained model is adapted into a one-step diffusion codec. To preserve the learned generative priors, we fine-tune the diffusion backbone using LoRA (Hu et al., 2022) with rank 32. With a batch size of 16 and learning rate of $10^{-4}$, we first train with $L_1$ and LPIPS losses for 200k steps, then incorporate PatchGAN adversarial loss for an additional 100k steps.

**Evaluation Protocol.** We construct an evaluation set of 1,000 images by randomly selecting one image per class from the ImageNet validation set. As shown in Figure 2, we report two FID metrics: (1) **Pre-Train FID** measures generation quality after Stage I by sampling 1,000 images and computing FID against the evaluation set; (2) **Codec FID** measures compression quality after Stage II using overlapped $64 \times 64$ patches following (Mentzer et al., 2020).

## A.2. Analysis on Diffusion Transformers in CoD

This subsection provides additional details for the attention analysis in Section 4, including visualization methodology, statistical analysis, and ablation configurations.

**Attention Map Visualization.** Figure 3 (Part 2) visualizes attention patterns in a pre-trained CoD model. For each DiT block, we compute the attention map by averaging attention scores across all heads for a given query position (the center point in the illustrated example). We further present more visualizations results covering additional query locations, timesteps, and input images in Figure 11. The local focus pattern is consistently observed across all tested configurations, validating our conclusion that global attention is largely redundant in CoD.

**Statistical Analysis.** Figure 3 (Part 3) presents quantitative analysis of attention patterns:

*Sub-figure (a):* We aggregate attention scores across all point-pairs, blocks, and heads at timestep $0.5T$, computing the weighted attention mass at each spatial distance. The results confirm that attention mass is heavily concentrated at short distances.

*Sub-figure (b):* We select the top-$K\%$ attention scores ($K\% \in \{1\%, 20\%, 50\%, 100\%\}$) from all heads within each block and compute the weighted average distance. This analysis is conducted at timestep $0.5T$.

*Sub-figure (c):* Similar to sub-figure (b), but we average across all blocks and evaluate at multiple timesteps. This reveals that local focus becomes more pronounced as noise decreases.

**Ablation Study Configuration.** Table 1 compares the performance of global attention, local window attention, and depth-wise convolution. For local attention, we use a window size of 3, i.e., each token calculates attention within a $3 \times 3$ window. For depth-wise convolution, we use $3 \times 3$ kernels. The multi-step pre-training follows the same pipeline as our main codec (Section 6 and Appendix A.3), with the exception that PatchGAN is used instead of DMD and projected GAN during Stage II fine-tuning.

## A.3. Real-Time Diffusion-Based Compression

This subsection details the training configuration for our proposed real-time codec, including dataset composition, hyperparameter settings, and computational requirements.

**Training Data.** Following CoD (Jia et al., 2025b), we curate a diverse training set comprising three public datasets: 9.3M images at $256 \times 256$ resolution from **ImageNet-21K** (Russakovsky et al., 2015), 1.7M images at $512 \times 512$ resolution from **OpenImages** (Kuznetsova et al., 2020), and 11.1M images at $512 \times 512$ resolution from **SA-1B** (Kirillov et al., 2023). This yields a total of 22M training images. For low-resolution training at $256 \times 256$, all images are resized accordingly.

**Hyperparameters.** In Equation 1, we set the loss weights as $\lambda_{\text{DMD}} = 2$ and $\lambda_{\text{GAN}} = 0.01$.

**Training Schedule.** We employ a progressive multi-stage

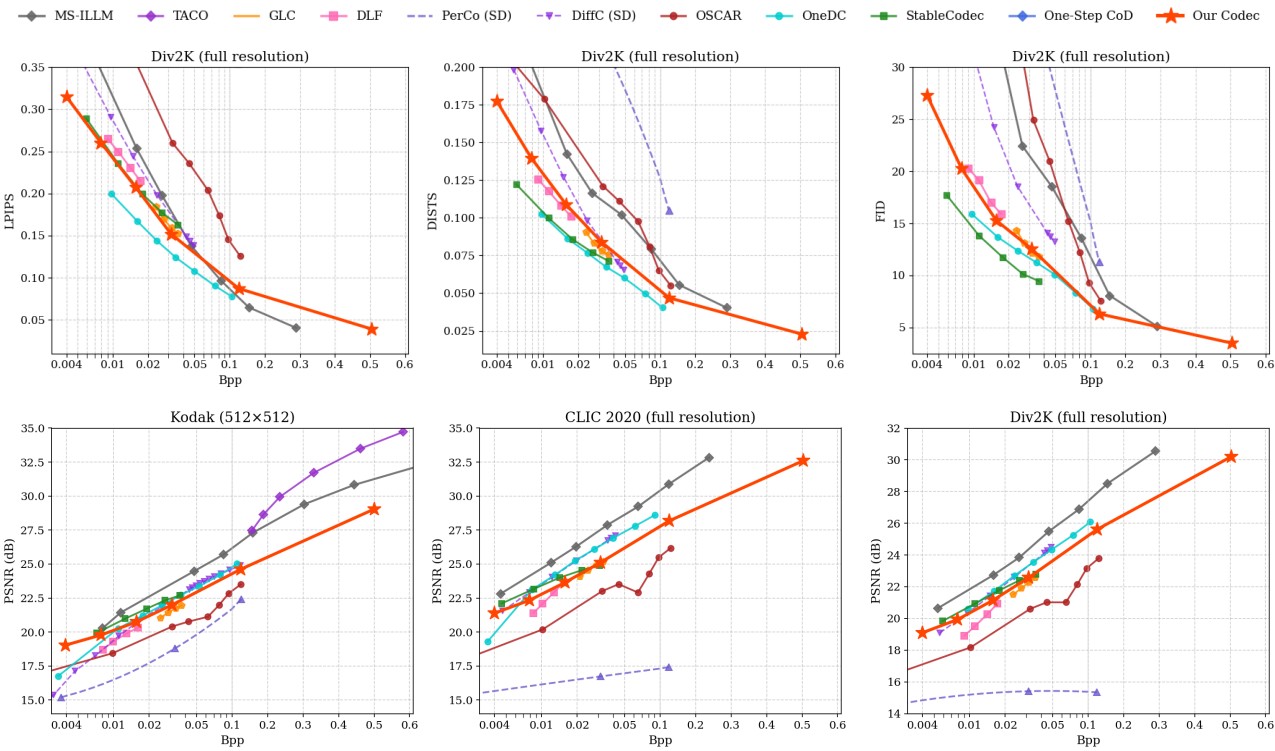

*Figure 9.* Rate-perception curves on Div2K and rate-distortion curves on all datasets.

| Stage | Image Resolution | BPP | #Images | Training Steps | Batch Size | Learning Rate | GPU hours (A100) |
|---|---|---|---|---|---|---|---|
| Low-Resolution Pre-Training | $256 \times 256$ | 0.0156 bpp | 22.1M | 600 K | $4 \times 32$ | $1 \times 10^{-4}$ | $4 \times 56$ |
| High-Resolution Pre-Training | $512 \times 512$ | 0.0039 bpp | 12.8 M | 100 K | $4 \times 16$ | $2 \times 10^{-5}$ | $4 \times 9$ |
| Unified Post-Training | $512 \times 512$ | 0.0039 bpp | 12.8 M | 50 K | $4 \times 16$ | $2 \times 10^{-5}$ | $4 \times 6$ |
| Reconstruction Fine-Tuning | $512 \times 512$ | Target bpp | 12.8 M | 200 K | $4 \times 4$ | $1 \times 10^{-4}$ | $4 \times 18$ |
| Dist. & Adv. Fine-Tuning | $512 \times 512$ | Target bpp | 12.8 M | 200 K | $4 \times 8$ | $1 \times 10^{-5}$ | $4 \times 43$ |

*Table 5.* Detailed configuration for each training stage. The complete pre-training pipeline requires 284 A100 GPU hours ($\approx$ 12 A100 days), while fine-tuning for each target bitrate requires 244 A100 GPU hours ($\approx$ 10 A100 days).

training strategy on 4 A100 GPUs. The detailed configuration for each stage is provided in Table 5. The complete pre-training pipeline requires 284 A100 GPU hours, while fine-tuning at each target bitrate requires an additional 244 A100 GPU hours.

## B. More Experimental Results

This section presents additional experimental results that complement the main paper, including extended rate-distortion/perception curves, high-resolution fine-tuning experiments, and qualitative visual comparisons.

### B.1. Rate-Distortion and Rate-Perception Curves

Figure 5 in the main paper presents rate-perception curves on Kodak ($512 \times 512$) and CLIC 2020 (full resolution) using LPIPS, DISTS, and FID metrics. Here, we provide extended

results:

**Additional Datasets.** Figure 9 presents results on Div2K (Agustsson & Timofte, 2017) along with rate-distortion curves (PSNR) across all datasets. Our codec achieves competitive PSNR performance compared to state-of-the-art one-step diffusion methods, demonstrating that perceptual optimization does not significantly compromise distortion metrics.

**Large Model Variant.** Figure 10 shows rate-perception and rate-distortion curves for our large codec variant with a 556M decoder. This scaled model achieves state-of-the-art FID scores on Kodak while maintaining competitive performance on other metrics. These results correspond to the large model data point in Figure 1.

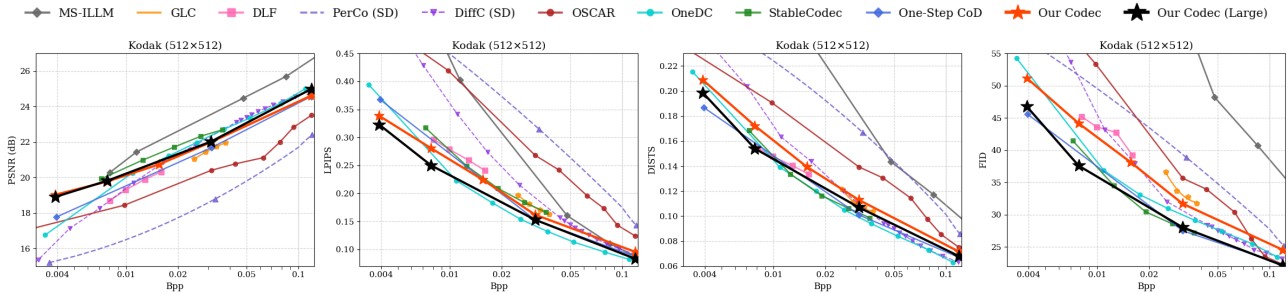

*Figure 10.* Rate-perception and rate-distortion curves for our large codec.

| One-Step Fine-Tune @ 0.0312 bpp | FID | FID w/ DMD |
|---|---|---|
| DiT → Global Attn. | 37.5 | 35.4 |
| + Replace Attn. by Conv. | 41.4 | 36.4 |
| + Improved Training | 38.9 | 35.2 |

*Table 6.* Effects of improve training of convolution networks.

## B.2. Other Results

**Improved training of convolution network**. In Table. 1, replacing global attention with convolution presents only marginal loss, which we attribute this as the optimization difficulty of depth-wise convolution networks. To demonstrate this, we further improve the CNN training by refining learning rate and extending the training schedule in Table. 6, where CNN outperforms DiT with improved training.

## B.3. Visual Results

Figure 12 presents additional visual comparisons between our codec and baseline methods on the Kodak dataset.

**Ultra-low bitrates.** Despite having substantially fewer parameters, our codec reconstructs images with high fidelity at ultra-low bitrates, such as 0.0039 bpp.

**High bitrates.** Most existing diffusion-based codecs are constrained by the latent capacity of VAEs, which limits their performance at high bitrates. In contrast, our codec supports high-quality compression at 0.5 bpp and consistently outperforms GAN-based codecs in this regime.

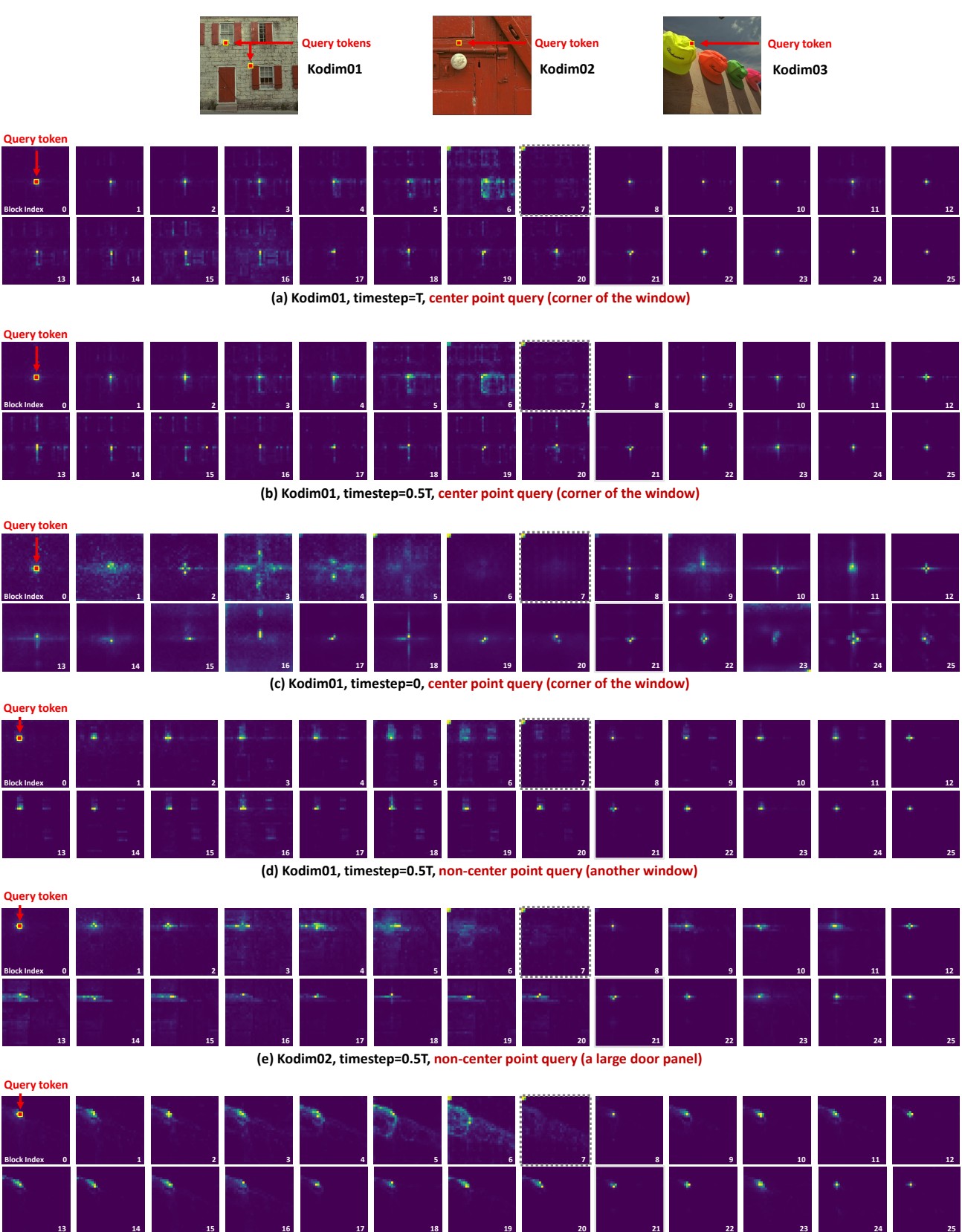

*Figure 11.* Additional attention map visualizations of DiT in CoD across different query locations, timesteps, and images. The observed local focus pattern is consistent across all examples.

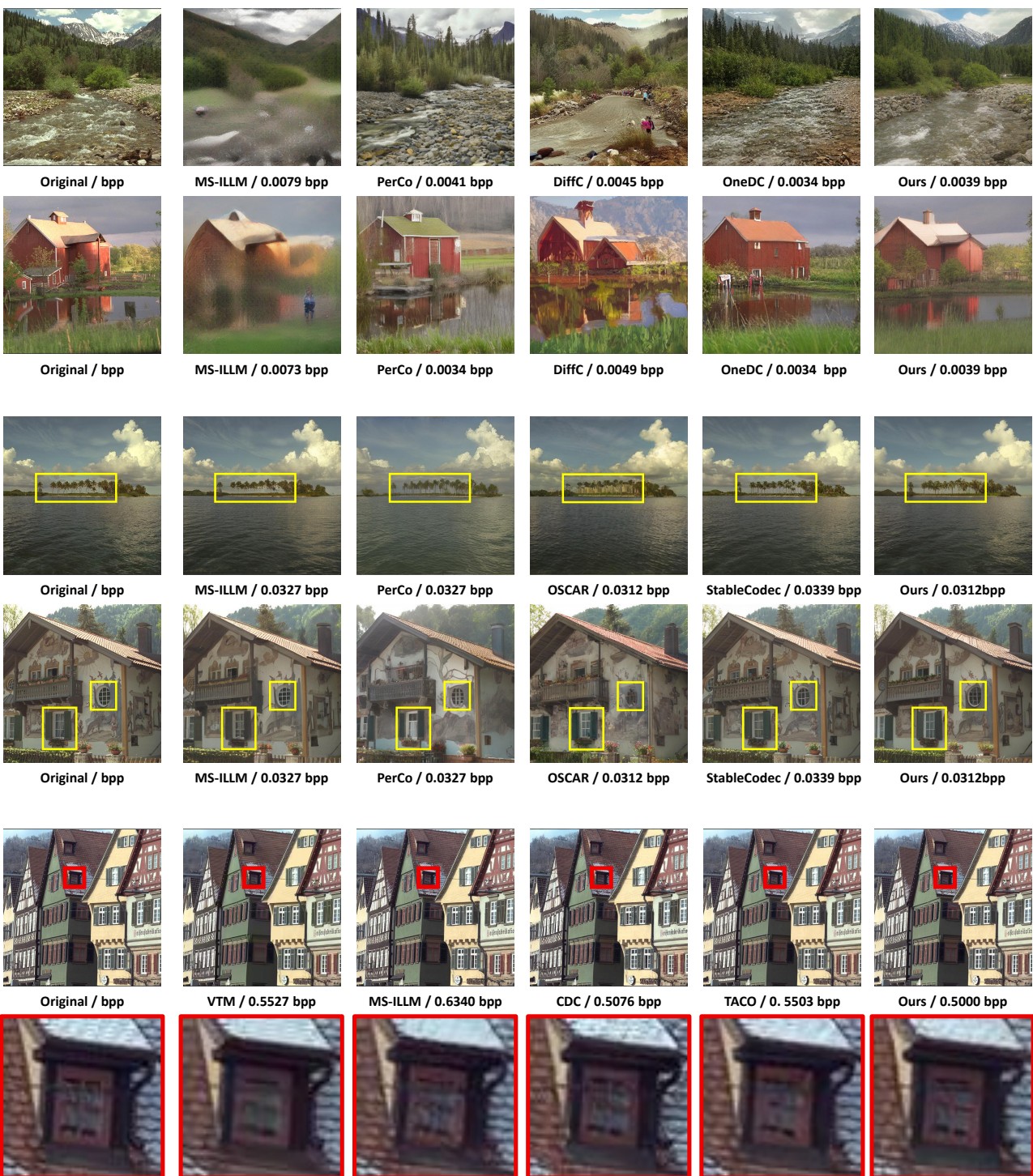

*Figure 12.* More visual comparison examples.

