# OpenReview forum: "CoD-Lite: Real-Time Diffusion-Based Generative Image Compression"
_ICML.cc/2026/Conference — ICML 2026 regular_

### Official Review · Reviewer_dSGB · 2026-03-06

**Soundness:** 3
**Presentation:** 3
**Significance:** 3
**Originality:** 3
**Overall Recommendation:** 4
**Confidence:** 3

**Summary:**

The authors propose a real-time, lightweight convolutional diffusion framework. Through systematic experimentation, the study reveals that scaling provides diminishing returns for lightweight models compared to larger ones. The authors present two primary insights: (1) providing information-dense conditioning is more effective for small-scale models than traditional generation-oriented pre-training, and (2) global attention in Transformer architectures tends to collapse into local focus during compression tasks. Based on these findings, they propose replacing Transformers with efficient convolutional layers. The resulting method achieves real-time 1080p encoding/decoding while outperforming existing diffusion-based methods in perceptual quality.

**Compliance With Llm Reviewing Policy:**

Affirmed.

**Final Justification:**

I have read the authors' response. Considering the incomplete results, such as the missing RD cost curves, combined with the comments from other reviewers, I tend to maintain my score.

**Key Questions For Authors:**

The paper identifies that 19 out of 26 layers exhibit attention locality. Does this characteristic fluctuate based on the bitrate? Specifically, does global attention become more critical at extremely low bitrates, where conditioning information is sparser and provides less structural guidance?

**Limitations:**

yes

**Strengths And Weaknesses:**

Strengths

* Real-time diffusion-based compression is highly significant for low-latency applications such as live video streaming and mobile communications.
* The paper provides a compelling quantitative and visual analysis justifying why lightweight diffusion models can move away from Transformers. The discovery of attention redundancy in the presence of strong compression priors is instructive and well-supported.
* The work achieves a breakthrough in the rate-distortion-perception trade-off, maintaining competitive generative quality while reaching speeds suitable for practical deployment.

Weaknesses

* Generative compression typically excels in perceptual metrics but often underperforms in traditional distortion metrics like PSNR or SSIM. The paper lacks a thorough discussion on how much reconstruction fidelity is sacrificed to achieve such high speed and generative quality.
* The paper moves directly to a new architecture. However, it lacks a discussion on whether similar real-time performance could be achieved by applying model compression techniques (e.g., pruning or quantization) to existing Transformer-based diffusion models.
* The first analysis regarding the diminishing returns of model scaling is primarily observed and concluded based on Transformer architectures. It remains unclear whether this scaling behavior is an inherent property of the task or if a CNN-based architecture might exhibit different scaling characteristics.

---

> ### Author Rebuttal · Authors · 2026-03-30
>
> We thank the reviewer for their careful review and valuable feedback. The rebuttal is provided below.
>
> ----
> ----
> ### **[W1] Discussion and Comparison on PSNR/SSIM Comparisons**
>
> We discuss the PSNR performance in Appendix B.1, where our codec achieves competitive results compared to SOTA diffusion-based codecs. Following the rate-distortion-perception trade-off theory, optimizing perceptual quality inherently incurs a PSNR drop (typically up to 3dB). Like all perceptual codecs, our method fall in this regime, trading PSNR for high realism.
>
> To clarify the exact distortion trade-off, we compare our codec against VTM, ELIC, and recent perceptual codecs at low (<0.01 bpp) and relatively high (0.03~0.05 bpp) bitrates in the table below. Our codec achieves comparable PSNR to other perceptual models and remains near the theoretical ~3 dB difference bound compared to MSE-optimized codecs (e.g., a ~3.4 dB drop vs. VTM at 30% fewer bits). We will include the complete RD curves in the final revision.
>
> | Method | Low bpp | PSNR / MS-SSIM | High bpp | PSNR / MS-SSIM |
> | :- | :- | :- | :- | :- |
> | **VTM** | - | - | 0.0447 | 25.4 dB / 0.85 |
> | **ELIC** | - | - | 0.0476 | 25.9 dB / 0.86 |
> | **MS-ILLM** | 0.0081 | 20.3 dB / 0.60 | 0.0479 | 24.4 dB / 0.84 |
> | **GLC** | - | - | 0.0335 | 21.7 dB / 0.76 |
> | **DLF** | 0.0081 | 18.7 dB / 0.54 | - | - |
> | **StableCodec** | 0.0072 | 19.9 dB / 0.62 | 0.0362 | 22.7 dB / 0.80 |
> | **OneDC** | 0.0034 | 16.7 dB / 0.44 | 0.0369 | 22.7 dB / 0.85 |
> | **Our Codec** | 0.0039 | 19.0 dB / 0.57 | 0.0312 | 22.0 dB / 0.78 |
>
>
>
> ----
> ### **[W2] Model Compression Techniques on DiTs**
>
> Model compression is indeed a promising direction for efficient networks.
>
> Regarding model pruning, the main idea is to reduce model size to enhance speed. However, we note that the main speed bottleneck of DiTs is not model size, but the $O(N^2)$ complexity of self-attention, which is not resolved simply by scaling down the parameter count. As shown in Table 1 of the main paper, we evaluated the speed of a lightweight diffusion DiT to demonstrate that merely reducing model size is insufficient for fast coding. We further extend this experiment to an even more extreme model size of ~10M parameters; as shown below, DiTs still lag significantly in decoding speed.
>
> | Dec. Speed on 1080p image \ Parameters | ~600M | ~200M | ~50M | ~10M |
> | :--- | :--- | :--- | :--- | :--- |
> | **DiT** | 517 ms | 414 ms | 331 ms | 180 ms |
> | **Our Codec (Conv)** | 66 ms | 33 ms | 24 ms | 12 ms |
>
> Model quantization is also a valuable avenue for future exploration. However, it still faces the $O(N^2)$ complexity of self-attention, and importantly, quantization can be equally applied to CNNs. While advanced techniques like attention pruning or sparse attention might mitigate the complexity issue, they introduce significant implementation overhead. Ultimately, convolutions are generally more efficient than sparse attention mechanisms. In this paper, we demonstrate that convolutions can be highly competitive with full attention for this task, suggesting that a CNN-based architecture is highly sufficient and effective without requiring complex attention approximations.
>
>
> ----
> ### **[W3] ImageNet experiments for CNNs**
>
> We conducted the requested experiment on our convolution-based diffusion codecs. Due to time and resource limitations during the rebuttal period, we could not execute the full training setups (which require extensive pre-training and fine-tuning across several ~700M parameter diffusion models). Instead, we reduced the training steps and compared: (1) training from scratch, (2) generation-oriented pre-training, and (3) compression-oriented pre-training. We measured the reconstruction FID of the fine-tuned codecs. As shown below, compression-oriented pre-training remains the primary source of gain for a smaller model, consistent with the scaling behavior we observed in DiTs.
>
> | Model / rFID | 700M | 34M |
> | :--- | :--- | :--- |
> | **From Scratch** | 24.3 | 30.9 |
> | **Generation Oriented** | 16.3 | 27.5 |
> | **Compression Oriented** | 13.5 | 24.3 |
>
> ----
> ### **[Q1] Attention Locality and bitrates**
>
> We agree that theoretically global attention should become more critical at lower bitrates where structural guidance is sparser. However, the attention analysis experiments in Section 4 were already conducted at an **extremely low 0.0039 bpp**, which approaches the lowest experimental bitrate for faithful reconstruction. Even at this extreme bitrate which is rarely encountered in practical applications, attention still exhibits strong locality. Therefore, at larger, more practical bitrates, the need for global attention becomes even less critical, further validating our architectural design.

---

> > ### Author Rebuttal · Reviewer_dSGB · 2026-04-03
> >
> > I thank the authors for their thorough response, which has addressed my major concerns. As a suggestion, it would be more appropriate to present these results alongside perceptual metrics.

---

> > > ### Author Response · Authors · 2026-04-03
> > >
> > > We are sincerely grateful for the reviewer's positive feedback and for the time dedicated to review our paper. We will present the new results as suggested in the revision.

---

### Official Review · Reviewer_DeF3 · 2026-03-08

**Soundness:** 3
**Presentation:** 3
**Significance:** 3
**Originality:** 2
**Overall Recommendation:** 4
**Confidence:** 3

**Summary:**

The paper addresses the limitations of existing compression methods: diffusion-based approaches are computationally expensive, while non-diffusion methods underperform on realistic image generation. The authors analyze the effectiveness of compression-oriented pre-trained diffusion models and evaluate the necessity of transformers. They highlight that compression-oriented pretraining works better than general diffusion pretraining, and attention mechanisms are not required.

Through empirical evaluation, the method performs well on a wide range of baselines across datasets, bitrate regimes, and perceptual metrics (FID, LPIPS, DISTS), achieving a balance of perceptual quality and computational cost. Qualitative comparisons further demonstrate the perceptual realism.

**Compliance With Llm Reviewing Policy:**

Affirmed.

**Final Justification:**

The additional details on the user study, along with the commitment to release code, address my main concerns. While the work is still largely empirical and engineering-focused, the results are better supported.

**Key Questions For Authors:**

Can the authors clarify why compression-oriented pretraining improves performance?

Have the authors considered conducting a human evaluation study to confirm that quantitative improvements translate to perceptual gains?

Are there any plans to release the implementation code to support reproducibility?

**Limitations:**

yes

**Strengths And Weaknesses:**

## Strengths

The problem is relevant to the machine learning community; they define it well in the introduction, and then motivate their design choices.

Methodology for evaluation is generally appropriate for the problem setting. Results are solid, showing improvements over many baselines in multiple metrics.

Presentation is clear, paper is well-positioned in related works.


## Weaknesses

The paper is a collection of engineering decisions to improve existing work, more of a technical report than a research paper.  Implementation code is not released, limiting reproducibility and follow-up research. There is limited key takeaway besides following some engeneering desicion they have made (convolution is fine, use CoD, and adjust hyperparameters).

No human evaluation study; it is unclear if quantitative improvements translate to perceptual gains.

Focus is primarily on empirical gains; originality of the approach is limited.

It is not explained why compression-oriented pretraining improves performance.

Figure 3 is hard to read, lables, legends are too small.

---

> ### Author Rebuttal · Authors · 2026-03-30
>
> ----
> ### **[W1.2 & Q3] Releasing the Code**
>
> We are fully committed to reproducibility and **we will release all the codes**, including pre-trained weights, training scripts and inference scripts, to ensure full reproducibility and facilitate follow-up research.
>
> ----
> ### **[W2 & Q2] Human Evaluation Study**
>
> We conducted a subjective user study, where 20 participants were asked to rate the reconstruction quality of images compressed at approximately 0.004 bpp on a scale of 1 (lowest) to 5 (highest), from which Mean Opinion Scores were derived. As shown below, our codec achieves superior perceptual quality compared to several state-of-the-art diffusion-based codecs.
>
> | Metric | PerCo | DiffC | OSCAR | OneDC | **Our Codec** |
> | :- | :- | :- | :- | :- | :- |
> | **User Score (1-5) ↑** | 3.5 | 2.1 | 2.3 | 3.8 | **4.0** |
>
> ----
> ### **[W3 & W1.1] Novelty and Originality Concerns**
>
> We respectfully disagree with the characterization of our work as a simple "collection of engineering decisions." As recognized by other reviewers, our proposed lightweight convolutional pixel-diffusion codec introduces a novel architecture (Reviewer `dSGB`: "The paper moves directly to a new architecture") that addresses a critical problem in neural media compression (Reviewers `dac3`, `dSGB`). Furthermore, we provide in-depth, principled analyses for each technique proposed:
> * **Convolutional Diffusion:** While existing diffusion models heavily rely on complex U-Net or DiT architectures, our analysis reveals that a massive receptive field is not strictly necessary for this task. Our observation that DiT attention rapidly collapses to a local focus provides strong empirical justification for our streamlined convolutional structure (Reviewers `Qz3J`, `dSGB` call this insight "brilliant" and an "airtight empirical basis"). This novel design choice is the primary driver of our model's high inference speed.
> * **Distillation-Guided Training:** We identify that distillation is the critical mechanism enabling a convolutional codec to perform competitively with transformer-based models. Leveraging a DiT as a teacher successfully compensates for the optimization difficulties inherent to depth-wise convolutions (noted by Reviewer `Qz3J`), offering an important methodological guideline for future CNN codec training.
> * **Compression-Oriented Diffusion Pre-training:** We are the first to thoroughly analyze the behavioral differences between generation-oriented and compression-oriented pre-training at smaller scales. We demonstrate that conditional entropy is the bottleneck for small models, explaining why generation-oriented pre-training fails while compression-oriented pre-training succeeds in this regime (noted by Reviewer `dac3`).
>
> In summary, our paper provides a collection of novel insights and scientifically grounded solutions. We identify fundamental bottlenecks, propose targeted architectural and training innovations, and establish strong, efficient baselines for future research.
>
>
>
> ----
> ### **[W4 & Q1] Why Compression-Oriented Pre-training Improves Performance**
>
> The general effectiveness of diffusion pre-training has been well-established by numerous diffusion-based codecs (e.g., PerCo, DiffEIC, StableCodec), demonstrating that generative priors significantly benefit fine-tuned codecs.
>
> Regrading compression-oriented diffusion (CoD), it has proven superior to traditional text-to-image generative pre-training for the following reasons:
> * **Information Density and Accuracy:** The text modality often lacks the capacity to describe precise, localized image details. At similar information densities, compression targets are much more structurally accurate. Consequently, learning compression-oriented diffusion provides a much stronger and more relevant prior for codec applications.
> * **Model Capacity Utilization:** Text-to-image diffusion models must expend significant capacity learning complex cross-modal alignments (text-to-vision). In contrast, compression-oriented diffusion focuses entirely on native image representations, utilizing the model's capacity much more efficiently.
>
> Furthermore, our analysis in Section 3 provides deeper insights into why this is particularly critical for **small models**:
> * The learning difficulty of a diffusion model is governed by both its parameter capacity and the condition information density.
> * For small models where capacity is strictly limited, the higher condition information density provided by compression-oriented pre-training simplifies the learning objective. This provides a significantly more effective pre-training phase, which directly translates to superior performance in the fine-tuned codec.
>
>
> ----
> ### **[W5] Issue on Figure 3**
>
> We will enlarge the labels, adjust the legends, and optimize the overall layout of Figure 3 to ensure clear readability in the revised manuscript.

---

> > ### Author Rebuttal · Reviewer_DeF3 · 2026-04-02
> >
> > Thank you for the rebuttal. Regarding the subjective user study, it would help to clarify a few points to properly assess the results. How many codec comparisons did each participant rate, and was the order of images or codecs randomized to avoid ordering bias? Also, did participants see the original uncompressed images as references, or were the ratings given without a reference? With 20 participants rating five codecs, providing information on the variability of scores would allow us to estimate a confidence interval and judge the statistical reliability of the reported perceptual improvements. These details are important to ensure the subjective gains are meaningful, and the methodology and results should be clearly described in the main text.

---

> > > ### Author Response · Authors · 2026-04-02
> > >
> > > We thank the reviewer for the constructive feedback. To clarify our subjective study:
> > >
> > > * Every participant rated all five codecs using a **Double-Stimulus** method, with the original uncompressed images provided as a reference.
> > > * Both the image sequences and the codec display orders were **fully randomized** to prevent ordering bias.
> > > * To demonstrate statistical reliability, we provide the **Standard Deviation (SD)** and **95% Confidence Interval (CI)** for the scores below:
> > >
> > > | Metric | PerCo | DiffC | OSCAR | OneDC | **Our Codec** |
> > > | :--- | :---: | :---: | :---: | :---: | :---: |
> > > | **User Score (1-5) ↑** | 3.50 ± 1.15 | 2.10 ± 0.79 | 2.30 ± 0.73 | 3.80 ± 0.89 | **4.00 ± 0.86** |
> > > | **95% CI** | [2.96, 4.04] | [1.73, 2.47] | [1.96, 2.64] | [3.38, 4.22] | **[3.60, 4.40]** |
> > >
> > > These details and the variability analysis will be clearly described in the revision.

---

### Official Review · Reviewer_dac3 · 2026-03-10

**Soundness:** 3
**Presentation:** 3
**Significance:** 3
**Originality:** 3
**Overall Recommendation:** 4
**Confidence:** 4

**Summary:**

This paper presents a real-time diffusion image codec. By replacing attention with convolutions and applying distillation, it achieves 42 FPS decoding at 1080p with perceptual quality comparable to SOTA methods, saving 85% bitrate over MS-ILLM.

**Compliance With Llm Reviewing Policy:**

Affirmed.

**Final Justification:**

The authors' rebuttal and additional experimental results are helpful and address my main concerns. I maintain my positive recommendation.

**Key Questions For Authors:**

1. Following W1, although the authors acknowledge the limited resolution setting, I am curious whether a toy experiment—comparing performance across 128×128, 256×256, 512×512 inputs—could offer some intuitions into how these modules scale with resolution.

2. The authors argue that global attention is redundant and can be replaced by convolutions. However, Table 1 shows that simply swapping attention for convolutions increases FID from 37.5 to 41.4, and distillation is needed to recover most of the gap. Can you explain the reason for this gap?

**Limitations:**

Yes. But as noted in Q1, a toy-model analysis on resolution generalization could further strengthen the discussion

**Strengths And Weaknesses:**

Strengths：

1. Addresses a critical problem in compression: this paper tackle the core bottleneck preventing diffusion models from being practically deployed in image compression—their latency. This work directly responds to an important and well-recognized need in the community.

2. Systematic empirical analysis: the authors compare 700M and 34M models under different pre-training conditions (Sec. 3), showing that generation-oriented pre-training fails for small models while compression-oriented pre-training succeeds. The controlled setup and clear takeaways make this analysis both rigorous and insightful.

3. Clear presentation: this paper is easy to follow and the step-by-step ablation roadmap (Fig. 7) further enhances readability.

Weakness:

1. Generalizability across resolutions is unclear: this paper's core claims—attention redundancy, one-step distillation feasibility, and the sufficiency of a 44M backbone—are established only at the tested scales (training at 512×512, testing up to 1920×1024). It remains uncertain whether these conclusions hold at higher resolutions (e.g., 4K/8K). see key questions.

---

> ### Author Rebuttal · Authors · 2026-03-30
>
> We thank the reviewer for their careful review and valuable feedback. The rebuttal is provided below.
>
> ----
> ----
> ### **[W1.1 & Q1] Resolution Robustness**
>
> Thank you for the valuable suggestion. We appreciate the opportunity to clarify our model's scalability across resolutions.
>
> - For **higher resolutions**, we tested our codec on 4K images and observed no block artifacts or spatial inconsistencies. Due to the text-only format of this rebuttal phase, we cannot attach visual results here, but we will include them in the revision. This robustness stems from the strong translation invariance of CNNs. The resolution limitation mentioned in our paper refers solely to a slight quantitative metric drop on high-resolution data (e.g., up to 2048×2048 in the CLIC2020 test set), and we believe fine-tuning our codec on higher resolution will eliminate this problem.
>
> - For **lower resolutions**, we compared our codec with SOTA baselines at lower resolutions (128×128 and 256×256). Surprisingly, our codec achieves vastly superior performance compared to all baselines at these scales. Particularly at 128×128, while other diffusion-based codecs achieve only marginal performance gains over MS-ILLM, our codec maintains substantial improvements. This furture proves our generalizability across resolutions.
>
> | BD-Rate on LPIPS / DISTS | Kodak 128×128 | Kodak 256×256 | Kodak 512×512 |
> | :- | :- | :- | :- |
> | MS-ILLM | 0 / 0 | 0 / 0 | 0 / 0 |
> | StableCodec | -31 / -38 | -44 / -70 | -36 / -77 |
> | OneDC | -13 / -11 | -41 / -72 | **-54 / -78** |
> | **Our Codec** | **-79 / -80** | **-62 / -75** | -40 / -70 |
>
> ----
> ### **[Q2] CNN Performance without DMD**
>
> Regarding the FID gap without DMD, we clarify that it is due to the **optimization difficulty** of CNNs on generative tasks. Comapred to attention, the depth-wise convolutions we employ are inherently more rigid, applying static filters across all spatial locations and channels. This makes them more challenging to optimize directly. However, as shown in the table below:
> - While CNN shows a higher FID than Transformer without DMD, it actually achieves **better DISTS score**, indicating superior structural texture preservation.
> - By refining the learning rate and extending the training schedule (CNN + improved training, as suggested by reviewer `Qz3J`), the FID improves to 38.9, closely matching the Transformer's 37.5. It indicates that more properly optimizing CNNs can ultimately reach performance of Transformer.
> - With DMD loss as a distribution-level optimization target, the CNN (35.2) ultimately outperforms the Transformer (35.4).
>
> | Method | DISTS | FID | FID w/ DMD |
> | :- | :- | :- | :- |
> | Transformer | 0.118 | 37.5 | 35.4 |
> | CNN | 0.115 | 41.4 | 36.4 |
> | CNN + improve training | 0.114 | 38.9 | 35.2 |

---

> > ### Author Rebuttal · Reviewer_dac3 · 2026-04-01
> >
> > Thanks for authors' reply. I have no further questions.

---

> > > ### Author Response · Authors · 2026-04-02
> > >
> > > We are sincerely grateful for the reviewer's positive feedback and for the time dedicated to review our paper.

---

### Official Review · Reviewer_Qz3J · 2026-03-11

**Soundness:** 3
**Presentation:** 3
**Significance:** 3
**Originality:** 3
**Overall Recommendation:** 5
**Confidence:** 3

**Summary:**

This paper addresses the severe inference latency bottleneck in current diffusion-based generative image compression methods by proposing a lightweight, real-time single-step convolutional diffusion image codec. Through empirical analysis, the authors reveal two core insights: (1) compression-oriented diffusion (CoD) pre-training provides crucial high-information conditioning for lightweight models, and (2) the attention mechanism in diffusion models for compression tasks primarily focuses locally, justifying the replacement of high-complexity $\mathcal{O}(N^2)$ Transformers (DiT) with lightweight depth-wise convolutions. Combined with Distribution Matching Distillation (DMD) and projected GAN techniques, the method achieves a decoding speed of 42 FPS at 1080p resolution on an A100 GPU (with only a 52M parameter decoder), while saving approximately 85% of the bitrate compared to MS-ILLM at a similar FID.

**Compliance With Llm Reviewing Policy:**

Affirmed.

**Final Justification:**

I recommend accepting this paper. The work presents a highly original and significant approach to real-time generative image compression by replacing DiT with depth-wise convolutions. The authors' rebuttal successfully resolved my main concerns regarding model optimization and real-world deployability by providing convincing consumer GPU benchmarks (e.g., RTX 4090) and extended training results. The rebuttal reinforced my positive assessment of the paper's soundness and clarity, and I advocate for its acceptance.

**Key Questions For Authors:**

1. Could you also provide 1080p latency results on a modern consumer-grade GPU (e.g., RTX 4090) or an edge NPU to better substantiate real-world deployability?
2. Please include complete PSNR and SSIM comparison curves against ELIC or VTM. The community needs to understand the exact distortion trade-off made to achieve these FID scores.
3. Since the CNN's performance degrades heavily without DMD (FID 41.4), does this imply depth-wise convolutions are fundamentally insufficient for single-step diffusion without a 700M teacher? Can the pure CNN recover its performance if trained significantly longer with just standard adversarial losses (no DMD)?
4. To dispel concerns about the limited receptive field of convolutions, could you include full-frame decoded results (with zoomed-in local patches) for 4K or extreme-aspect-ratio images?

**Limitations:**

yes

**Strengths And Weaknesses:**

strength:
1. The observation that DiT attention rapidly collapses to local focus (Layers 8-25) after the initial feature alignment stage (Layers 0-7) is brilliant, providing an airtight empirical basis for replacing DiT with convolutions.
1. The motivation is convincing. The observation that attention behaves heterogeneously provides an airtight empirical basis for replacing DiT with convolutions.
2. The approach combines high-information compression-guided pre-training with DMD distillation and adversarial fine-tuning. Leveraging a 700M-parameter DiT as a teacher to guide a 40M lightweight CNN successfully compensates for the optimization difficulties inherent to depth-wise convolutions.

weakness:
1. The 52M lightweight decoder comes at a massive computational cost. Training the 700M teacher and the student requires approximately 528 A100 GPU hours (nearly 22 days). Furthermore, Table 1 shows that without DMD distillation, the pure CNN's FID degrades severely to 41.4. This suggests the CNN lacks endogenous generative capability and relies heavily on over-fitting/memorizing the large teacher's outputs.
2. It remains unclear whether decoding images significantly larger than the training resolution (e.g., 4K) will produce block artifacts or spatial inconsistencies. The paper lacks qualitative demonstrations to ease this concern.
3. The PSNR curves in Figure 8 indicate that its pixel-level distortion performance lags behind several baselines at extremely low bitrates. This may limit the performance of the method.

---

> ### Author Rebuttal · Authors · 2026-03-30
>
> We thank the reviewer for their careful review and valuable feedback. The rebuttal is provided below.
>
> ----
> ----
> ### **[W1.1] Training Cost**
>
> We respectfully clarify that our training cost is actually lower than recent perceptual codecs. The training cost of perceptual codecs typically consists of two parts:
> - Base model pre-training (e.g., diffusion models).
> - Fine-tuning the base model into a codec.
>
> As shown below, our total training cost is lower than previous SOTA methods, especially considering the pretrain cost. We agree that further reducing the training cost is an important aspect, and we will explore it in the future though more efficient training strategy and hyper-parameters.
>
> | Method | Base Model Training Cost | Codec Training Cost |
> | :- | :- | :- |
> | **DLF** (ICCV '25) | TiTok pretrain: 197 A100 days | 56 A100 days |
> | **StableCodec** (ICCV '25) | SD + SD-Turbo pretrain: 6250 + 24 A100 days | ~7 A100 days |
> | **OneDC** (NeurIPS '25) | SD + DMD2 pretrain: 6250 + 69 A100 days | 72 A100 days |
> | **Our NVC** | CoD teacher + student pretrain: **20 + 12 A100 days** | **10 A100 days** |
>
>
> ----
> ### **[W1.2 & Q3] CNN Performance without DMD**
>
> We clarify that the FID gap is an **optimization issue** rather than a lack of generative capability. Comapred to attention, the depth-wise convolutions we employ are inherently more rigid, applying static filters across all spatial locations and channels. This makes them more challenging to optimize directly. However, as shown in the table below:
> - While CNN shows a higher FID, it actually achieves **better DISTS score**, indicating superior structural texture preservation.
> - As suggested, by refining the learning rate and extending the training schedule (CNN + improved training), the FID improves to 38.9, closely matching the Transformer's 37.5.
> - With DMD loss as a distribution-level optimization target, the CNN (35.2) ultimately outperforms the Transformer (35.4).
>
> | Method | DISTS | FID | FID w/ DMD |
> | :- | :- | :- | :- |
> | Transformer | 0.118 | 37.5 | 35.4 |
> | CNN | 0.115 | 41.4 | 36.4 |
> | CNN + improve training | 0.114 | 38.9 | 35.2 |
>
>
> ----
> ### **[W2 & Q4] Decoding at 4K Resolution and Receptive Field**
>
> Due to the text-only rebuttal format, we cannot attach visual results here. However, we will include full-frame 4K decoded images and zoomed-in local patches in the revision.
>
> When testing our codec on 4K images, we **do not observe any block artifacts or spatial inconsistencies**. This robustness is a direct benefit of the **translation invariance** inherent in CNNs. The "resolution limitation" mentioned in our paper refers strictly to a minor quantitative metric drop on high-resolution data (e.g., upto 2048x2048 in CLIC2020), not to any structural failures or obvious visual defects.
>
>
> ----
> ### **[W3 & Q2] Distortion Trade-off and PSNR/SSIM Comparisons**
>
> Following the rate-distortion-perception trade-off theory, optimizing perceptual quality inherently incurs a PSNR drop (typically up to 3dB). Like all perceptual codecs, our method fall in this regime. To achieve higher PSNR, there are several potential drections:
> 1) Use MSE (which directly maximizes PSNR) instead of L1 loss.
> 2) Use scalar quantization with entropy model to replace VQ-baesd information bottleneck. We found that it yields about a 1.5dB PSNR gain, which we leave for future work.
>
> To clarify the exact distortion trade-off, we compare our codec against VTM, ELIC, and recent perceptual codecs at low (<0.01 bpp) and high (0.03~0.05 bpp) bitrates below. Our codec achieves comparable PSNR to other perceptual models and falls within the theoretical ~3dB difference bound compared to MSE-optimized codecs (e.g., ~3.4dB drop vs. VTM at 30% fewer bits). We will include the complete RD curves in the revision.
>
> | Method | Low bpp | PSNR / MS-SSIM | High bpp | PSNR / MS-SSIM |
> | :- | :- | :- | :- | :- |
> | **VTM (MSE-opt)** | - | - | 0.0447 | 25.4 dB / 0.85 |
> | **ELIC (MSE-opt)** | - | - | 0.0476 | 25.9 dB / 0.86 |
> | **MS-ILLM** | 0.0081 | 20.3 dB / 0.60 | 0.0479 | 24.4 dB / 0.84 |
> | **GLC** | - | - | 0.0335 | 21.7 dB / 0.76 |
> | **DLF** | 0.0081 | 18.7 dB / 0.54 | - | - |
> | **StableCodec** | 0.0072 | 19.9 dB / 0.62 | 0.0362 | 22.7 dB / 0.80 |
> | **OneDC** | 0.0034 | 16.7 dB / 0.44 | 0.0369 | 22.7 dB / 0.85 |
> | **Our Codec** | 0.0039 | 19.0 dB / 0.57 | 0.0312 | 22.0 dB / 0.78 |
>
>
> ----
> ### **[Q1] 1080p Latency on Consumer GPUs**
> To substantiate real-world deployability, we benchmarked our NVC on consumer-grade GPUs (RTX 4090 and RTX 2080Ti) in addition to the A100. As shown below, on an RTX 4090, our model encodes 1080p images at 59 FPS and decodes at 48 FPS, which is even faster than on the A100. Even on an older RTX 2080Ti, it achieves a practical encoding speed of 21 FPS. We will add these hardware benchmarks to the final manuscript.
>
> | Enc. / Dec. Time | NVIDIA A100 | RTX 4090 | RTX 2080Ti |
> | :- | :- | :- | :- |
> | **1920×1024** | 16 ms / 24 ms | 17 ms / **21 ms** | 46 ms / 47 ms |

---

> > ### Author Rebuttal · Reviewer_Qz3J · 2026-04-02
> >
> > Dear Authors,
> >
> > Thank you for the comprehensive rebuttal. The additional hardware benchmarks (RTX 4090/2080Ti) and the extended CNN training results have fully addressed my concerns regarding real-world deployability and model optimization. Your explanation of the training costs and the PSNR trade-offs is also clear and convincing.
> >
> > Please make sure to include these new metrics and the 4K qualitative results in your final manuscript. I maintain my positive stance on this work and raising my score to Accept.

---

> > > ### Author Response · Authors · 2026-04-02
> > >
> > > We are sincerely grateful for the reviewer's positive feedback and for the time dedicated to re-evaluating our manuscript. We will include the mentioned results in the revision.

---

### Decision · Program_Chairs · 2026-04-30

**Decision:**

Accept (regular)

**Comment:**

All reviewers agreed this paper tackles the core challeng preventing diffusion models from practical and efficient deployment in image compression with systematic empirical analysis and clear presentation. Compression-oriented pretraining yields consistently better performance for lightweight models, and DiT attention rapidly collapses to local focus, replacing transformers with convolutions.

While the initial concerns, including real-world deployability via various benchmarks, resolution generalizability, PSNR rate-distortion trade-off clarity, and statistical reliability of the human evaluation study, are reasonble, the main concerns are adequately addressed in rebuttal.

Therefore, I recommend acceptance, and encourage the authors to incorporate the suggested revisions in the camera-ready version.